# Protective Effects of *Dipterocarpus tuberculatus* in Blue Light-Induced Macular Degeneration in A2E-Laden ARPE19 Cells and Retina of Balb/c Mice

**DOI:** 10.3390/antiox12020329

**Published:** 2023-01-31

**Authors:** Su Jin Lee, Yu Jeong Roh, Ji Eun Kim, You Jeong Jin, Hee Jin Song, Ayun Seol, So Hae Park, Bounleuane Douangdeuane, Onevilay Souliya, Sun Il Choi, Dae Youn Hwang

**Affiliations:** 1Department of Biomaterials Science (BK21 FOUR Program), Life and Industry Convergence Research Institute, College of Natural Resources and Life Science, Pusan National University, Miryang 50463, Republic of Korea; 2Institute of Traditional Medicine, Ministry of Health, Vientiane 0103, Laos; 3School of Pharmacy, Henan University, Kaifeng 475004, China; 4Longevity & Wellbeing Research Center, Laboratory Animals Resources Center, College of Natural Resources and Life Science, Pusan National University, Miryang 50463, Republic of Korea

**Keywords:** age-related macular degeneration, *D. tuberculatus*, A2E, antioxidant, oxidative stress, inflammatory response

## Abstract

Natural products with significant antioxidant activity have been receiving attention as one of the treatment strategies to prevent age-related macular degeneration (AMD). Reactive oxygen intermediates (ROI) including oxo-N-retinylidene-N-retinylethanolamine (oxo-A2E) and singlet oxygen-induced damage, are believed to be one of the major causes of the development of AMD. To investigate the therapeutic effects of methanol extracts of *Dipterocarpus tuberculatus* Roxb. (MED) against blue light (BL)-caused macular degeneration, alterations in the antioxidant activity, apoptosis pathway, neovascularization, inflammatory response, and retinal degeneration were analyzed in A2E-laden ARPE19 cells and Balb/c mice after exposure of BL. Seven bioactive components, including 2α-hydroxyursolic acid, ε-viniferin, asiatic acid, bergenin, ellagic acid, gallic acid and oleanolic acid, were detected in MED. MED exhibited high DPPH and ABTS free radical scavenging activity. BL-induced increases in intracellular reactive oxygen species (ROS) production and nitric oxide (NO) concentration were suppressed by MED treatment. A significant recovery of antioxidant capacity by an increase in superoxide dismutase enzyme (SOD) activity, SOD expression levels, and nuclear factor erythroid 2–related factor 2 (NRF2) expression were detected as results of MED treatment effects. The activation of the apoptosis pathway, the expression of neovascular proteins, cyclooxygenase-2 (COX-2)-induced inducible nitric oxide synthase (iNOS) mediated pathway, inflammasome activation, and expression of inflammatory cytokines was remarkably inhibited in the MED treated group compared to the Vehicle-treated group in the AMD cell model. Furthermore, MED displayed protective effects in BL-induced retinal degeneration through improvement in the thickness of the whole retina, outer nuclear layer (ONL), inner nuclear layer (INL), and photoreceptor layer (PL) in Balb/c mice. Taken together, these results indicate that MED exhibits protective effects in BL-induced retinal degeneration and has the potential in the future to be developed as a treatment option for dry AMD with atrophy of retinal pigment epithelial (RPE) cells.

## 1. Introduction

Age-related macular degeneration (AMD) is characterized by the deposition of lipofuscin and drusen and the development of abnormal blood vessels in the retina and macula, which eventually lead to the total loss of central vision [1,2]. Specifically, an important component of lipofuscin, N-retinylidene-N-retinylethanolamine (A2E), which is produced from the reaction between phosphatidylethanolamine and vitamin A, plays an important role in the pathological progression of AMD. A2E accumulates in the retinal pigment epithelial (RPE) cells during the aging process [3]. Accumulated A2E can cause cellular injury due to its amphiphilic structure and photoreactivity and this is believed to be the pathophysiological process leading to atrophic AMD [4]. In the RPE cells, A2E contributes to the production of reactive oxygen species (ROS) and several by-products after exposure to blue light (BL) [5,6]. These products induce chronic inflammation through the activation of the complement system, cause lysosomal damage, increase cellular damage by destroying DNA and disrupt the intracellular autophagy process [7,8,9]. Several natural products with high antioxidant activities are being used for treatment and protection against A2E-induced damage in AMD. Significant protective effects against oxidative damage, apoptosis, and inflammatory response were detected in cells and animal models with retinal disorders and AMD phenotypes after treatment with *Aronia melanocarpa* [10], bilberry [11], Shihu Yeguang Pill [12], triphala [13], *Solanum melongena* L. [14], bilberry and lingonberry [15,16], *Vaccinium uliginosum* [17] and *Spirulina maxima* [18].

*Dipterocarpus tuberculatus* Roxb. is a species of flowering plant widely distributed in Southeast Asia including Bangladesh, Cambodia, Laos, Myanmar, Thailand, and Vietnam [19]. The therapeutic effects of this plant as an anti-inflammatory, anti-photoaging, and in promoting osseointegration have been reported recently. Ethanol (EtOH, 90%) extracts of this plant suppressed the lipopolysaccharide (LPS)-mediated inflammatory response in RAW264.7 cells and improved acute inflammatory symptoms in EtOH/HCl-induced gastric lesions of ICR mice through the regulation of 3′-phosphoinositide–dependent kinase 1/Nuclear factor kappa-light-chain-enhancer of activated B cells (PDK1/NF-κB), MyD88/TIR-domain-containing adaptor protein inducing IFN-β (TRIF-mediated) activated protein (AP)-1 and AP-1 signaling pathways [20,21]. Additionally, some significant anti-photoaging effects of the methanol extract of *Dipterocarpus tuberculatus* Roxb. (MED) in apoptosis, the cell cycle, extracellular matrix structure, and inflammation were detected in ultraviolet (UV)-irradiated primary normal human dermal fibroblasts (NHDF) cells and nude mice [22]. MED, coated on the surface of a titanium plate stimulated the proliferation and cell adhesion of MG63 cells as well as new bone formation and regeneration in tibia implantation models [23]. However, no study to date has elucidated the protective effects and the mechanism of action of *Dipterocarpus tuberculatus* Roxb. in retinal cells and animals with AMD phenotypes.

In this study, we investigated the protective effects of MED and the mechanisms involved in A2E-laden ARPE-19 cells and Balb/c mice after exposure to blue light (BL) to evaluate its potential as a treatment for AMD.

## 2. Materials and Methods

### 2.1. Preparation and Extraction of MED

The lyophilized sample of MED (FBM 213-075) was supplied from the International Biological Material Research Center of the Korea Research Institutes of Bioscience and Biotechnology (KRIBB, Daejeon, Korea). To collect the extract solution of MED, a ground dried powder was first prepared from dry stem samples of *Dipterocarpus tuberculatus* Roxb. using a blender. After mixing a 1:10 ratio of MED powder and methanol solution, they were subjected to two steps: (1) sonication for 15 min and (2) incubation for 2 h at room temperature, and these steps were repeated 10 times per day for 3 days. After that, the extracted solution was filtered through a filter with a 0.4 µm pore size and lyophilized using a Rotary Evaporator (EYELA, Bohemia, NY, USA). The lyophilized MED samples were deposited as voucher specimens of *Dipterocarpus tuberculatus* Roxb. (WP-20-001) at the functional materials bank of the Wellbeing RIS Center, Pusan National University (PNU). These lyophilized samples were prepared by dissolving them in dimethyl sulfoxide solution (DMSO; Duchefa Biochemie, Haarlem, The Netherlands) before treatment.

### 2.2. Determination of Bioactive Compounds in MED

The bioactive compounds in MED were determined in accordance with methods laid out in a previous study [22]. Seven bioactive compounds were identified in MED namely 2α-hydroxyursolic acid, ε-viniferin, asiatic acid, bergenin, ellagic acid, gallic acid, and oleanolic acid, using ultra-high-performance liquid chromatography (UHPLC)-electrospray ionization (ESI)-tandem mass spectrometry (MS/MS) (UHPLC-ESI-MS) analysis. Among these, three compounds with high content were quantified using the liquid chromatography-mass spectrometry (LC-MS) quantification method. The observed peak area ratios were calculated from the detected peak area in total running time (Appendix A).

### 2.3. Free radical Scavenging Activity of MED

The scavenging activity against 2,2-diphenyl-1-picrylhydrazyl (DPPH) radicals was determined using a previously described method [22,24]. Briefly, various solutions of MED (1 to 1000 µg/mL) were mixed with 0.1 mM DPPH (Sigma-Aldrich Co., St. Louis, MO, USA). After incubation at room temperature for 30 min, the absorbance of each mixture was determined at 517 nm using a VersaMax^TM^ microplate reader (Molecular Devices, Sunnyvale, CA, USA). Finally, the scavenging activity of MED against DPPH radicals was expressed as the reduction percent in absorbance. The IC_50_ value was defined as the MED concentration that gives a 50% decrease in the scavenging activity against DPPH radicals.

Additionally, the 2,2′-azono-bis-3-ethylbenzthiazoline-6-sulphonate (ABTS) radical scavenging activity was determined using an ABTS decolorization assay as described in a previous study [25]. A total of 25 μL of eleven different concentrations of MED (1 µg to 500 µg/mL) were mixed with 250 μL of ABTS working solution and incubated at room temperature for 4 min. Their absorbance was read at 734 nm in a UV-visible (UV–VIS) spectrophotometer (Thermo Fisher Scientific Inc., Wilmington, DE, USA). The data have been presented as ascorbic acid equivalent (Sigma-Aldrich Co., St. Louis, MO, USA), which was used as a standard.

### 2.4. Synthesis and Purification of A2E

To synthesize A2E, all-trans-retinal (100 mg, 352 μmol), ethanolamine (9.5 mg, 155 μmol) in ethanol (3 mL), and acetic acid (9.3 µL, 155 μmol) were mixed by vigorous vortexing for 2 min. The mixture was incubated for 48 h under dark conditions. Subsequently, the mixture was concentrated at 20 °C using a nitrogen evaporator (Biotage, TurboVap-LV, USA). An A2E sample was separated and purified using silica gel and HPLC. Separation on a Sep-Pak C18 cartridge (WAT023635, Waters, Milford, MA, USA) was performed using a step gradient elution with 5:95 MeOH/CH_2_Cl_2_, 5:95 MeOH/CH_2_Cl_2_, and 8:92:0.001 MeOH/CH_2_Cl_2_/TFA. Additionally, the fractions were further purified using a multiple preparative HPLC (LC-forte/R, YMC Co., Kyoto, Japan) with a YMC-Triart Prep C18 column (250 mm × 10.0 mm, 10 μm). The purity of the A2E was analyzed using the gradient HPLC method with a YMC-Triart C18 column (4.6 mm × 250 mm, 5 µm) (Appendix A).

### 2.5. Cell Culture and Cell Viabilities Assay

ARPE19 cells, derived from the human RPE, were sourced from the American Type Culture Collection (ATCC), (Manassas, VA, USA), and grown in an incubator maintained with a 5% CO_2_, 95% atmosphere and 37 °C temperature using Dulbecco’s Modified Eagle Medium (DMEM, Welgene, Gyeongsan, Korea) containing 10% fetal bovine serum (FBS).

To determine the optimal concentration of MED, ARPE19 cells (3 × 10^4^ cells) were briefly seeded into each well of a 96-well plate. When the cells were up to 70–80% confluence, they were treated with various concentrations of MED (50, 100, 200, and 400 μM) for 24 h. Supernatants were discarded after incubation for 24 h, followed by the addition of fresh DMEM (200 μL) and 3-(4,5-dimethylthiazol-2-yl)-2,5-diphenyltetrazolium bromide (MTT) solution (20 mg/mL and 50 μL) to each well. After incubation for 4 h, the formazan precipitates in the cells were dissolved in DMSO (Duchefa Biochemie, Haarlem, The Netherlands), and their absorbance was determined at 570 nm using a VersaMax^TM^ microplate reader (Molecular Devices, San Jose, CA, USA). Based on the above results, the optimal concentration of MED was determined at 50, 100, and 200 μg/m (Appendix A).

Next, we determined an optimal dosage for A2E at specific BL irradiation levels to prepare the AMD cell model. To achieve this, ARPE19 cells cultured in the same manner as above were treated with various concentrations of A2E (5, 10, 20, 40, and 80 μM) for 24 h and irradiated with BL (430 nm, 6000 Lux) (SL-S2500, S tech LED, Gyeonggido, Republic of Korea) for 10 min. After further incubation for 24 h, the viability of cells was measured by an MTT assay as described in the section above. The non-irradiated group (Non treated group) did not receive any BL radiation, but the Vehicle + A2E treated group received only BL irradiation without A2E loading. Based on the above results, the optimal concentration of A2E was determined at 20 μM (Appendix A).

To assess the protective effects of MED on the oxidative stress-induced cell death caused by BL in A2E-laden ARPE-19 cells, they were briefly divided into two groups; Not treatment group (Non treated group) and the A2E + BL treated group. The A2E + BL treated group was further classified into the following five groups; Vehicle (DMSO) treated group (Vehicle + A2E + BL treated group), Vitamin C (100 μM, positive control) treated group (Po + A2E + BL treated group), low concentration (50 μg/mL) MED treated group (LMED + A2E + BL treated group), medium concentration (100 μg/mL) MED treated group (MMED + A2E + BL treated group) and high concentration (200 μg/mL) MED treated group (HMED + A2E + BL treated group). Vitamin C was used as a positive control because it has been proven to have high antioxidant activity and a protective effect on macular degeneration [26]. When ARPE19 cells were up to 70–80% confluence, they were treated with three different concentrations of MED (50, 100, and 200 μg/mL), Vitamin C or DMSO of the same volume for 24 h, and then treated with 20 μM of A2E for 24 h. Subsequently, the cells were irradiated by BL (430 nm, 6000 Lux) for 10 min. After further incubation for 24 h, the viability of cells was measured by an MTT assay as described in the section above. Additionally, these cells were further subjected to a real-time-quantitative polymerase chain reaction (RT-qPCR), the Western blot test, and apoptotic cell analyses.

### 2.6. Analysis of Apoptotic Cells

The distribution of apoptotic and live cells was analyzed using a Muse^TM^ Annexin V and Dead Cell Kit (Millipore Co., Billerica, MA, USA). After harvesting ARPE19 cells treated with MED, A2E, or BL, they (1 × 10^4^ cells/mL) were mixed with the Muse^TM^ Annexin V and Dead Cell Kit (Millipore Co., Billerica, MA, USA) reaction reagent and subsequently incubated for 20 min. These cells were analyzed using the Muse^TM^ Cell Analyzer (Millipore Co., Billerica, MA, USA). Briefly, they were gated based on their size, their population was classified into four populations: live cells [Annexin V (−) and 7-AAD (−) population], early apoptotic cells [Annexin V (+) and 7-AAD (−) population], late apoptotic cells [Annexin V (+) and 7-AAD (+) population], mostly nuclear debris [Annexin V (+) and 7-AAD (+) population].

### 2.7. Determination of Intracellular ROS Levels

The intracellular ROS levels determined using 2′,7′-dichlorofluorescein diacetate staining analysis (DCFH-DA; Sigma-Aldrich Co., St. Louis, MO, USA) based on this probe can be deacetylated by intracellular esterase to nonfluorescent 2′,7′-dichlorodihydrofluorescein (DCFH). This chemical reacted with intracellular hydrogen peroxide or other oxidizing ROS and was consequently changed into fluorescent 2′,7′-dichlorofluorescein (DCF). ARPE19 cells were subsequently treated with MED, A2E, or BL as described previously. After the final incubation, the cells were treated with 10 µM DCF-DA for 30 min at 37 °C and subsequently washed with 1× PBS. The green fluorescence in ARPE19 cells was observed using a fluorescent microscope (Evos m5000, Thermo Fisher Scientific Inc., Waltham, MA, USA) at 400× magnification.

### 2.8. Analysis of Nitric Oxide (NO) Concentrations

The level of nitrite in the ARPE19 cells was determined as an indicator of NO production using the Griess reagent as in a previous study [22]. Briefly, the culture supernatants were collected from subset groups of ARPE19 cells after subsequent treatment with MED, A2E, or BL. These supernatants (100 μL) were mixed with 100 μL of modified Griess reagent (Invitrogen Co., Carlsbad, CA, USA) in 96-well plates. After incubation for 5 min, the absorbance at 540 nm was detected using a Versamax^TM^ microplate reader (Molecular Devices, San Jose, CA, USA).

### 2.9. Analysis of Superoxide Dismutase (SOD) Activity

The SOD activity in the ARPE19 cells was assessed using a SOD assay kit (Dojindo Molecular Technologies Inc., Rockville, MD, USA) according to the procedure suggested by the manufacturer. Briefly, ARPE19 cells were harvested from the subset groups after treatment with MED, A2E, or BL. They were lysed by repetitive freezing and thawing in 100 μL of 1× PBS, and then their lysates were collected by centrifugation at 5000× *g* for 5 min. After diluting with 1× PBS in seven ratios (1/1, 1/2, 1/22, 1/23, 1/24, 1/25, and 1/26), these samples (20 μL each) were divided into individual wells of a 96-well plate. The solution in each well was thoroughly mixed with water-soluble tetrazolium salt-1 (WST-1) working solution (200 μL) and enzyme working solution (20 μL). After incubation at 37 °C for 20 min, their absorbance was measured at 450 nm using a VersaMax^TM^ microplate reader (Molecular Devices, San Jose, CA, USA). Finally, the SOD activity of each group was determined using the following equation:SOD activity (inhibition rate %) = [(A blank 1 − A blank 3) − (A sample − A blank 2)]/(A blank 1 − A blank 3) × 100(1)
where, A blank 1, 2, and 3 represented the absorbance level of blanks 1, 2, and 3, and ‘A sample’ indicated the absorbance level of sample.

### 2.10. Western Blot Analysis

The total cellular proteins were prepared from ARPE19 cells, using the Pro-Prep Protein Extraction Solution (Intron Biotechnology Inc., Seongnam, Republic of Korea). After lysis of ARPE19 cells, total proteins were harvested, and their concentration was sequentially determined using a SMARTTM Bicinchoninic Acid Protein Assay Kit (Thermo Fisher Scientific Inc., Waltham, MA, USA). Then, proteins bound to the nitrocellulose membranes were incubated with the specific primary antibodies (Appendix A) overnight at 4 °C. The intensity for each protein was analyzed on the membrane, which developed with an Amersham ECL Select Western Blotting detection reagent (GE Healthcare, Little Chalfont, UK) using the Fusion Solo-2 (Vilber, San Leandro, Collégien, France). Finally, the density of each protein was quantified using the AlphaView Program (Cell Biosciences Inc., Santa Clara, CA, USA).

### 2.11. RT-qPCR Analysis

The relative quantities of inflammatory cytokines (TNF-α, IL-6, IL-1β and NF-κB) mRNA were determined by RT-qPCR analyses [27,28]. After isolating total RNA molecules using RNA Bee solution (Tet-Test Inc., Friendswood, TX, USA), complement DNA (cDNA) was synthesized with reverse transcriptase (Superscript II, Thermo Fisher Scientific Inc., Waltham, MA, USA). Four specific genes were amplified with 2× Power SYBR Green (Toyobo Co., Osaka, Japan) [29] using specific primers (Appendix A). Finally, the expression of each gene was quantified at a relative level to that of the actin (housekeeping gene) by comparing the Cts at a constant fluorescence intensity [30].

### 2.12. Experimental Design for Animal Study

The study protocol for the AMD animal model was carefully reviewed and approved by the Pusan National University-Institutional Animal Care and Use Committee (PNU-IACUC) (Approval Number PNU-2022-0103). Male Balb/c mice (5 weeks old) were provided from Samtako BioKorea Inc. (Osan, Korea), and bred in the barrier facility of the PNU-Laboratory Animal Resources Center (PNU-LARC), accredited by the AAALAC International (Unit Number; 001525) and the Korea Food and Drug Administration (KFDA) (Unit Number; 000231). They were supplied, ad libitum, with filtered tap water and a standard irradiated chow diet (Samtako BioKorea Co., Osanm, Korea). All mice were maintained in a specific pathogen-free (SPF) state, strict regulation of the light cycle, constant temperature (22 ± 2 °C) and relative humidity (50 ± 10%).

The therapeutic effects of MED in the AMD model were analyzed as described in a previous study [14]. Briefly, the 7-week-old Balb/c mice (Male, *n* = 24) were allocated to one of two groups; a non-irradiated group (Non treated group, *n* = 8) and a BL irradiated group (*n* = 16). The latter group was further assigned into four groups; the Vehicle-treated group (1× PBS, Vehicle + BL treated group, *n* = 8), the low-concentration MED-treated group (100 mg/kg, LMED + BL treated group, *n* = 8), or high-concentration MED-treated group (200 mg/kg, HMED + BL treated group, *n* = 8). However, only the Vehicle treated group was not assigned to the Non treated group based on the 3R principle of PNU-IACUC to reduce the number of animals because there was no significant difference between the Non treated group and the Vehicle treated group in the preliminary experiment. After adapting to the dark conditions for three days, they were orally administrated with the same volume of vehicle solution (1× PBS), or MED solution (100 mg/kg or 200 mg/kg) once a day for 4 days in dark cages. The dosages for MED treatment used in the AMD model were decided based on the results from previous research on the anti-AMD effects of natural products [11,31] and the therapeutic effects of *Dipterocarpus tuberculatus* Roxb. [21,22]. At 24 h after the final administration, they were exposed to BL for 2 h and subsequently bred for 24 h under dark conditions. Subsequently, all mice were sacrificed using CO_2_ and their eye samples were collected for histopathological analyses.

### 2.13. Histopathological Analysis

After the collection of the eyeballs from the Blab/c mice, the entire eye was fixed in 10% formalin and then embedded in paraffin wax. After sectioning into 4 μm thick slices, they were stained using hematoxylin and eosin solution (H&E, Sigma-Aldrich Co., St. Louis, MO, USA). The histological features on these sections were observed by optical microscopy, after which the thickness of the whole retina, outer segment of the photoreceptors (OS), the outer nuclear layer (ONL), inner nuclear layer (INL), and inner plexiform layer (IPL) for photoreceptor degeneration in the retina was observed using the Leica Application Suite (Leica Microsystems, Glattbrugg, Switzerland)

### 2.14. Immunohistochemical (IHC) Staining Analysis

The tissue distribution of nuclear factor erythroid 2–related factor 2 (Nrf2), cyclooxygenase-2 (COX-2), and inducible nitric oxide synthase (iNOS) proteins were detected by IHC as described in a previous study [32]. After deparaffinization of tissue sections, anti-Nrf2 (Abcam, Cambridge, UK), anti-COX-2 (Cell Signaling Technology Inc., Danvers, MA, USA), anti-iNOS (Thermo Fisher Scientific Inc., Waltham, MA, USA) and goat alkaline phosphatase (AP) conjugated anti-rabbit IgG (1:200, Thermo Fisher Scientific Inc., Waltham, MA, USA), antibodies were sequentially treated onto these sections. Finally, the distribution of each protein in the retina was detected using stable diaminobenzidine (DAB) (Invitrogen Co., Waltham, MA, USA) and evaluated using the Leica Application Suite (Leica Microsystems).

### 2.15. Statistical Analysis

One-way ANOVA was used to determine the statistical significance between the Vehicle + A2E + BL (or Vehicle + BL) treated group and MED + A2E + BL (or MED + BL) treated group, and only a *p* value less than 0.05 was reported as statistically significant. All values in the results are represented as the means ± standard deviation (SD).

## 3. Results

### 3.1. Bioactive Compounds and Antioxidative Activity of MED

Firstly, we evaluated the antioxidative activity of MED based on its bioactive compound composition to predict its potential as a therapeutic drug for AMD. Seven bioactive compounds were detected in MED using HPLC analyses. Among these, asiatic acid was present in the highest amounts, followed by 2α-hydroxyursolic acid, oleanolic acid, bergenin, ε-viniferin, gallic acid, and ellagic acid (Figure 1a and Appendix A). Additionally, inhibitory activity against DPPH and ABTS radicals was significantly increased at 1–1000 µg/mL of MED, and the IC_50_ value for DPPH and ABTS radicals was determined to be 7.48 µg/mL and 505.12 µg/mL (Figure 1b,c). Specifically, the IC_50_ value for DPPH radicals was remarkably higher than that of the ABTS radicals (Figure 1b,c). These results suggest that MED shows strong antioxidative activity and has the potential for application as a protective drug for AMD.

### 3.2. Protective Effects of MED on the Oxidative Stress Caused by A2E + BL Treatment in ARP19 Cells

To investigate whether the treatment of MED can protect against the oxidative stress caused by AMD, changes in intracellular ROS production and NO concentration were measured in MED +A2E + BL treated ARPE19 cells. The number of DCF-stained cells representing intracellular ROS was remarkably higher in the Vehicle + A2E + BL treated group compared to the Non treated group. However, the above high values were significantly decreased by 18.2%, 50%, and 66.7% in the LMED + A2E + BL, MMED + A2E + BL, and HMED + A2E + BL treated groups, respectively, compared to the Vehicle + A2E + BL treated group (Figure 2a,b). Additionally, a similar recovery pattern with a decrease rate of 8.6%, 20%, and 33.3%, respectively, was detected in the NO concentrations in the same groups although the rate of decrease was greater in the DCF stained cells (Figure 2c). Thus, these results indicate that MED may contribute to the decrease in oxidative stress in A2E + BL treated ARPE19 cells through the suppression of ROS and NO production.

### 3.3. Protective Effects of MED against the Loss of Antioxidant Capacity Caused by A2E + BL Treatment in ARPE19 Cells

To examine whether the treatment of MED can protect against the reduction in antioxidative capacity in AMD, we measured the alterations in the SOD activity and expression, and the Nrf2 expression in MED + A2E + BL treated ARPE19 cells. Three parameters related to the antioxidative capacity exhibited a similar pattern in the three MED-treated groups. SOD activity was lower by 37% in the Vehicle + A2E + BL treated group compared to the Non treated group. However, this level was remarkably enhanced only in the HMED + A2E + BL treated group compared to the Vehicle + A2E + BL treated group (Figure 3a). However, a remarkable recovery in the expression level of antioxidative proteins was observed in all the MED-treated groups. The expression level of SOD and the Nrf2 proteins were decreased in the Vehicle + A2E + BL treated group compared to the Non treated group. However, the above decreased values were significantly increased in the three MED + A2E + BL treated groups. Especially, the levels of these proteins were higher than that of the Non treated group in the MMED + A2E + BL and HMED + A2E + BL treated groups (Figure 3b). Taken together, the data of the present study indicate that the protective effects of MED may be tightly associated with the recovery of antioxidative capacity loss caused by A2E + BL treatment in ARPE19 cells.

### 3.4. Protective Effects of MED against Cell Death Caused by A2E + BL Treatment in ARPE19 Cells

To examine whether the treatment of MED can protect against cell death in AMD, the alterations of the cell viability were first analyzed in the MED + A2E + BL treated ARPE19 cells. A significant decrease in cell viability was successfully induced in the A2E + BL treated ARPE19 cells, and these levels were remarkably protected in a dose-dependent manner in the MED + A2E + BL treated group (Figure 4). Additionally, these cells were stained with Annexin V and PI to analyze the distribution of apoptotic cells. The total number of apoptotic cells remarkably increased 7.5-fold in the Vehicle + A2E + BL treated group compared to the Non treated group. However, these enhanced values gradually decreased in the three MED + A2E + BL treated groups, although the highest rate of decrease was detected in the HMED + A2E + BL treated group (Figure 5). Moreover, the alterations in the number of apoptotic cells were well reflected in the expression of proteins responsible for the regulation of apoptosis. The increased levels of the Bax/Bcl-2 ratio and the cleaved Cas-3/Cas-3 ratio in the A2E + BL treated cells, significantly decreased with MED pretreatment. Bax/Bcl-2 expressions decreased in a dose-dependent change in the MED-treated groups (Figure 6). Therefore, the above results suggest that the protective effect of MED on A2E + BL-induced cell death may be related to the suppression of apoptosis in ARPE19 cells.

### 3.5. Protective Effects of MED on the Regulation of Angiogenesis Caused by A2E + BL Treatment in ARPE19 Cells

To investigate the therapeutic effect of MED on the regulation of angiogenesis caused by AMD, the alteration in the expression levels of matrix metalloproteinase (MMP) 2, 9 and vascular endothelial growth factor (VEGF) proteins in the MED + A2E + BL treated ARPE19 cells were analyzed. Treatment with MED led to a decrease in the expression levels of the three proteins, but there was a clear difference in the pattern of decrease. The expression level of MMP2 was significantly reduced equally in all MED-treated groups compared to the Vehicle + A2E + BL-treated group. However, the levels of MMP9 expression were significantly reduced in only the HMED + A2E + BL treated group. The decrease in expression was seen in both the LMED and MMED + A2E + BL treated groups; however, this decrease was not statistically significant. A significant dose-dependent decrease was seen in the expression of the VEGF proteins (Figure 7). These results suggest that MED may protect against the dysregulation of angiogenesis caused by A2E + BL treatment through the suppression of MMPs and VEGF expressions in ARPE19 cells.

### 3.6. Protective Effect of MED against the Inflammatory Response Caused by A2E + BL Treatment in ARPE19 Cells

To investigate the protective effects of MED against AMD-induced inflammatory response, alterations in the expression of the iNOS-induced COX-2 mediated pathway and inflammasome were measured in the MED + A2E + BL treated ARPE19 cells. The expression of iNOS and COX-2 were greater in the Vehicle + A2E + BL treated group than in the control group. The decrease in the expression of these proteins was dose-dependent in the MED-treated groups; however, their rate was different (Figure 8). Additionally, the protective effect of MED on the regulation of iNOS and COX-2 was well reflected in the inflammasome activation. The expression levels of three key members including NLR family pyrin domain containing 3 (NLRP3), apoptosis-associated speck-like protein containing a CARD (ASC), and the cleaved of Cas-1 proteins, were higher in the Vehicle + A2E + BL treated group than the Non treated group. However, the above-increased values were significantly decreased in the LMED, MMED and HMED + A2E + BL treated groups compared to the Vehicle + A2E + BL treated group. Especially, the expression levels of NLRP3 and ASC in the MED + A2E + BL treated groups were lower than that of the Non treated group, while the cleaved Cas-1/Cas-1 ratio was completely recovered in the Non treated group (Figure 9). Furthermore, the protective effects of MED on the regulation of iNOS-induced COX-2 mediated pathway and inflammasome were well reflected in the regulation of inflammatory cytokine expressions. The MED + A2E + BL treated groups exhibited a significant decrease in the mRNA levels of TNF-α, IL-6, IL-1β and NF-κB compared to the Vehicle + A2E + BL treated group (Figure 10). Therefore, the above results for inflammatory responses suggest that MED can protect against the increased inflammatory response seen in A2E + BL treated cells through the regulation of the iNOS-induced COX-2 mediated pathway and NLRP3 inflammasome activation in ARPE19 cells.

### 3.7. Protective Effect of MED against the Photoreceptor Degranulation Caused by BL in the Retina of Balb/c Mice

Finally, we investigated the protective effects of MED against photoreceptor degeneration in the retina of the AMD mice model to confirm the protective effects of MED detected in the A2E + BL treated ARPE19 cells. To achieve this, the alterations of the thickness of the whole retina, OS, ONL, and INL were measured in the retinas of the Balb/c mice treated with MED + BL. As shown in Figure 11, the retinal thickness was lower in the Vehicle + BL treated group than in the Non treated group. However, the thickness was significantly increased in the LMED + BL and HMED + BL treated groups compared to the Vehicle + BL treated group. Additionally, a similar pattern of increase was detected in the thickness of the OS, ONL, and INL although the thickness of the IPL was maintained constant. Furthermore, the expression of antioxidative proteins and iNOS-induced COX-2 mediated pathway-related proteins were analyzed in the retinas of BL-exposed Balb/c mice to investigate whether the protective effect of MED against retinal degeneration has an association with the alteration of the antioxidative activity and inflammatory response. The tissue density of the Nrf2 proteins was lower in the retinas of Vehicle + BL treated mice than in the Non treated group. This level gradually recovered in the LMED + BL and HMED + BL treated groups although it did not recover to the level of the Non treated group (Figure 12a). A reverse pattern was detected in the COX-2 and iNOS-stained retinal tissues. The increase in their density in the Vehicle + BL treated group was remarkably decreased after the MED treatment (Figure 12b,c). Therefore, these results suggest that MED administration can protect against the retinal degeneration of BL-exposed Balb/c mice through the regulation of antioxidative activity and the inflammatory response.

## 4. Discussion

Photochemical damage is induced by exposure of the retina to high-energy radiation [4]. During this process, the phagocytosis of the outer segment of photoreceptors and the production of superoxide anions are remarkably increased, leading to photo-oxidative stress in the retina [6,33]. Therefore, the administration of ROS scavengers can be considered one of the therapeutic strategies to protect or delay the progression of early AMD [34]. As part of a study aimed at identifying novel natural products with protective effects against AMD, we investigated the therapeutic effects and mechanism of action of MED in ARPE19 cells and Balb/c mice with AMD phenotypes. The results of our study provide novel scientific evidence that MED treatment may contribute to the protection against BL-induced retinal damage through the regulation of oxidative stress, apoptosis, neovascularization, and inflammatory response.

In the present study, the antioxidant properties of *Dipterocarpus tuberculatus* Roxb. are an important scientific and logical reason for its selection for evaluation as a potential therapeutic option for the treatment of AMD. As shown in Figure 1, Figure 2 and Figure 3, MED showed high free radical scavenging activity, ROS suppressive activity and NO production, and stimulatory activity for SOD and Nrf2 expressions. Similar activities of other natural products have been reported in a few previous studies. The ethanol extract of *Dipterocarpus tuberculatus* Roxb. leaves and twigs suppressed the release of NO and prostaglandin E2 (PGE2) from LPS-stimulated RAW264.7 cells through the inhibition of COX-2 and iNOS transcriptions as well as the NF-κB signaling pathway [20]. NO concentrations, SOD activity, and Nrf2 expressions improved significantly in UV-irradiated NHDF cells after treatment with MED [22]. Studies also show that natural products with good antioxidative activity slow down the progression of AMD. A2E photo-oxidation-induced damages of ARPE19 cells significantly improved after treatment with procyanidins B2, *Arctium Lappa* L., *Prunella vulgaris* var. L., *Solanum melongena* L. through the suppression of oxidative stress and high antioxidative activity [14,35,36,37]. Our study adds to the current body of evidence that suggests that natural products with high antioxidant activity can be considered potential treatments for a variety of diseases that occur due to oxidative damage.

Furthermore, in our study, asiatic acid, 2α-hydroxyursolic acid and ellagic acid are reported as the principal bioactive compounds as shown in Figure 1. They have high antioxidant activity and several biological functions even as a single compound. Asiatic acid is widely distributed in many fruits and vegetables and has various therapeutic effects on oxidative stress, inflammation and fulminant hepatic failure [38]. Similar to asiatic acid, 2α-hydroxyursolic acid has been discovered in a variety of medicinal herbs and fruits [39]. This compound shows also various pharmacological effects including antidiabetic activity, anti-inflammation, anti-obesity, anti-atherosclerosis, and anticancer [39,40,41,42,43]. Ellagic acid with high antioxidative activity is produced as a second metabolite through the hydrolysis of ellagitannins in various plants [44]. It has recently received a lot of attention because of its potential in the treatment of human diseases including diabetes, cancer, cardiovascular disease, and neurodegenerative diseases [45,46,47,48]. However, we believe that the protective effects of MED against AMD may be related to the synergistic effect of three compounds rather than the effect of a single component, although these effects include the effects of other unknown compounds.

Lipofuscin granules contain A2E and its oxidized forms (A2E-ox and A2E-2ox) accumulate in the retinal pigment epithelium with age, and this leads to macular degeneration [49,50]. During this process, the cumulative damage to the retina is accompanied by an increase in apoptosis, oxidative stress, and inflammation [51]. Based on the above scientific evidence, an in vitro model for AMD was established in ARPE19 cells laden with A2E and exposed to BL [49]. This model has been used to investigate the therapeutic effect of various natural products on AMD. Grape skin polyphenols, *Vaccinium uliginosum* L. extract, and procyanidins B2 significantly inhibited photo-oxidation-induced apoptosis in ARPE19 cells. Additionally, this inhibition involved the recoveries of the Bc1-2/Bax ratio, Cas-3, and Cas-9 cleavage, and recovery from endoplasmic reticulum (ER) stress [35,52,53]. Polyphenol-enriched *Vaccinium uliginosum* and norbixin reduced cell death, inhibited A2E accumulation, and resulted in improved photoprotection in ARPE19 cells and the primary culture of retinal cells, while the extract of *Curcuma longa* L. and curcuminoids protected against photo-oxidative damages and apoptosis in ARPE19 cells [17,54,55]. Similar effects of A2E accumulation, cell death, apoptosis, and cytokine expression were detected in ARPE-19 cells after treatment with *Arctium lappa* L., *Prunella vulgaris var* L., *Solanum melongena* L., and *Centella asiatica* [14,36,37,56]. In the present study, we investigated the therapeutic effects and the action mechanism of MED in A2E-laden ARPE19 cells after exposure to BL. MED treatment protected the cells by suppressing ROS and NO production, restoring antioxidant activity, suppressing the inflammatory response, and inhibiting apoptosis and angiogenesis. These results are consistent with previous research findings which investigated the therapeutic effects of various natural products in mammalian cells with AMD phenotypes. Therefore, our results provide the first evidence of the novel therapeutic function of MED against BL-induced macular degeneration in A2E-laden ARPE19 cells.

Meanwhile, the Nrf2 protein is well known as one of the important transcription factors that regulate the transcription of genes related to xenobiotics response and oxidative stress response to maintain cellular hemostasis [57]. Additionally, these responses involve many enzymes for detoxification including phase I, II and II drugs as well as the elimination of pro-antioxidants such as glutathione synthetase (GSS), SOD and catalase [58]. Recently, the regulatory mechanism of Nrf2 has been investigated in RPE cells by mimicking the onset of different eye diseases. Nrf2 overexpression improves the morphological structure of RPE cells and survival rate in mouse models of retinal degeneration through the regulation of multiple oxidative defense pathways and glutathione pathways [59]. Furthermore, the expression level of Nrf2 increased in the diabetic retinopathy conditions and is significantly recovered by treatment of polyunsaturated docosahexaenoic acid (DHA) in RPE cells [60]. A similar activation in the Nrf2 signaling pathway was induced in ARPE19 cells after treatment with hydrogen peroxide. This activation was ameliorated with idebenone, a ubiquinone short-chain synthetic analog which has high antioxidant properties [61]. The activation of Nrf2 and transcriptional upregulation of its downstream genes were induced by tertbutyl hydroperoxide (t-BHP)-mediated oxidative stress in human RPE cells although these responses were attenuated by pretreatment of *Ginkgo biloba* extracts (GBE) [62]. In the present study, the expression level of Nrf2 was increased in A2E-laden ARPE19 cells after treatment with BL. Therefore, the results of the present study suggest that the action mechanism of Nrf2 may be considered a common mechanism involved in oxidative stress response triggered by various chemical and physical agents including hydrogen peroxide, t-BHP and BL as well as mimicking the condition of eye diseases such as diabetic retinopathy and retinitis pigmentosa. Furthermore, it is believed that this mechanism of Nrf2 has the potential to be useful for evaluating the efficacy of antioxidants.

Finally, the protective effects and action mechanism of natural products against AMD in retinal epithelial cells were verified in the experimental animal model with retinal damage although these cases are very rare. Among them, most of this damage in Balb/c mice was induced by exposure to BL at 10,000 Lux for 1 h/day for 2 weeks [14,17,36,54]. However, only one AMD model with an increase in the photoreceptor thickness and ONL was produced by intraperitoneal injection of N-methyl-N-nitrosourea (MNU) at a concentration of 50 mg/kg [56]. In the AMD models, the retinal cells recovered from the damage with the administration of several natural products. The polyphenol-enriched *Vaccinium uliginosum* L. fraction inhibited the decrease in ONL thickness and nuclei, while the administration of norbixin reduced the A2E accumulation in the retina of Abca4^-/-^Rdh8^-/-^ mice [17,54]. Similar improvement effects were observed in AMD model animals after treatment with *Arctium lappa* L., *Prunella vulgaris var.* L., and *Solanum melongena* L. [14,36,37]. Moreover, the thickness of the photoreceptors, ONL, and nuclei numbers in MNU-induced AMD model animals recovered when treated with *Centella asiatica* [56]. The results seen with MED administration on the retina of BL-exposed Balb/c mice were also similar. The thickness of the OS, ONL, and INL significantly improved after the administration of MED. The present study demonstrated the protective effects of MED against photo-oxidative damages in the retina of AMD animal models. These results suggest a new therapeutic role for MED that has not been investigated previously.

## 5. Conclusions

In the present study, we attempted to demonstrate the novel therapeutic effects of MED against AMD using a mammalian cell and an experimental animal model. To achieve this objective, the antioxidant activity of MED as well as changes in antioxidant capacity, anti-apoptosis, anti-angiogenesis, and anti-inflammatory response in A2E-laden ARPE19 cells after exposure to BL were investigated. Additionally, these effects were further assessed in the retinas of BL-exposed Balb/c mice. Our results provided the first scientific evidence that the high antioxidant activity of MED has the potential to prevent AMD and relieve symptoms related to the condition (Figure 13). However, our study had some limitations in that it did not directly compare all the results obtained in vitro to those at all layer levels of the retina due to restrictions on the collection of eye tissue samples. Moreover, the lack of any mechanism analyses in relation to lysosomal damage and disruption of the autophagy process in MED + A2E + BL treated ARPE19 cells and MED + BL treated mice should be considered as a drawback of our study. Therefore, further research involving animal models will be needed on some processes of cell damage to understand the mechanisms of action of MED and components.

## Figures and Tables

**Figure 1 antioxidants-12-00329-f001:**
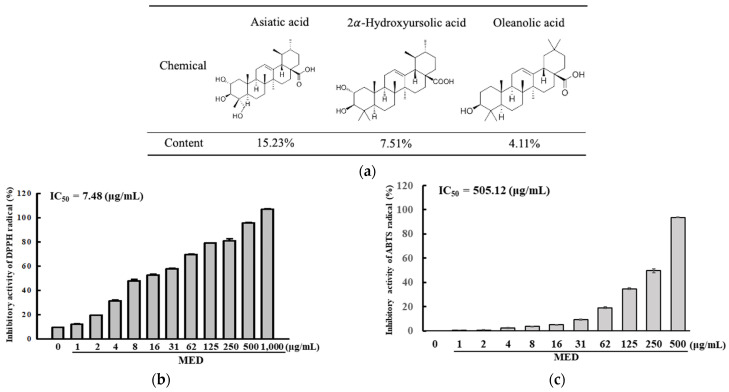
Main components and free radical scavenging activity of MED. (**a**) Concentration of asiatic acid, 2α-hydroxyursolic acid and oleanolic acid in MED. (**b**) DPPH radical scavenging activity of MED. This activity was measured in the concentration ranges from 1 to 1000 μg/mL of MED. Scavenging activity analysis for DPPH radicals was conducted with three MED samples. (**c**) ABTS radical scavenging activity of MED. This activity was measured in the concentration ranges from 1 to 500 μg/mL of MED. Scavenging activity analysis for ABTS radicals was conducted with three MED samples. All values in the results are represented as the means ± standard deviation (SD). Abbreviations: DPPH, 2,2-diphenyl-1-picrylhydrazyl; IC_50_, half maximum inhibitory concentration; ABTS, 2,2′-azino-bis (3-ethylbenzothiazoline-6-sulfonic acid); MED, methanol extracts of *Dipterocarpus tuberculatus* Roxb.

**Figure 2 antioxidants-12-00329-f002:**
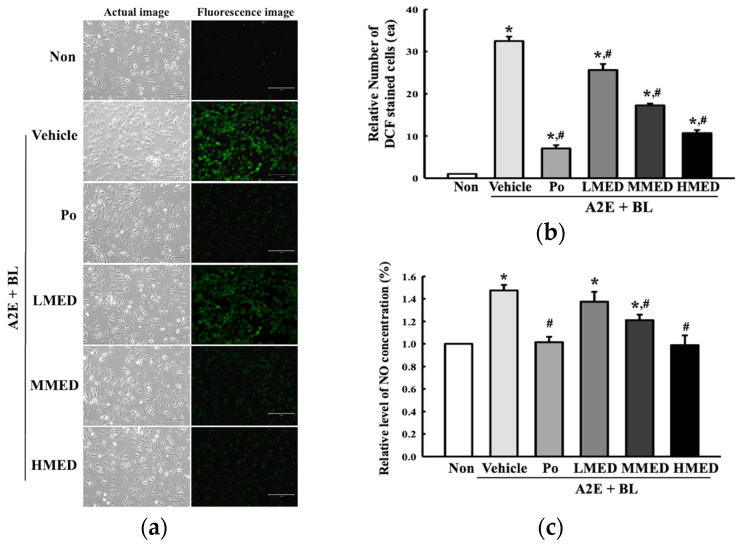
Detection of ROS and NO concentration in MED + A2E + BL treated ARPE19 cells. (**a**) Fluorescence image of DCF stained cells. The morphology of cells was detected using an optical and fluorescent microscope at 200× magnification. (**b**) Number of DCF stained cells. (**c**) NO concentration. NO concentration in the culture supernatants was determined using Griess reagent. DCFH-DA staining and NO assay were conducted using three wells per group, and these assays for each sample were analyzed in duplicates. All values in results are represented as the means ± standard deviation (SD). * indicated statically significance compared to the Non treated group, while ^#^ indicated statically significance compared to the Vehicle + A2E + BL treated group. Abbreviations: ROS, reactive oxygen species; NO, Nitric oxide; DCFH-DA, 20,70-Dichlorofluorescin diacetate; MED, methanol extracts of *Dipterocarpus tuberculatus* Roxb.; A2E, N-retinylidene-N-retinylethanolamine; BL, Blue light.

**Figure 3 antioxidants-12-00329-f003:**
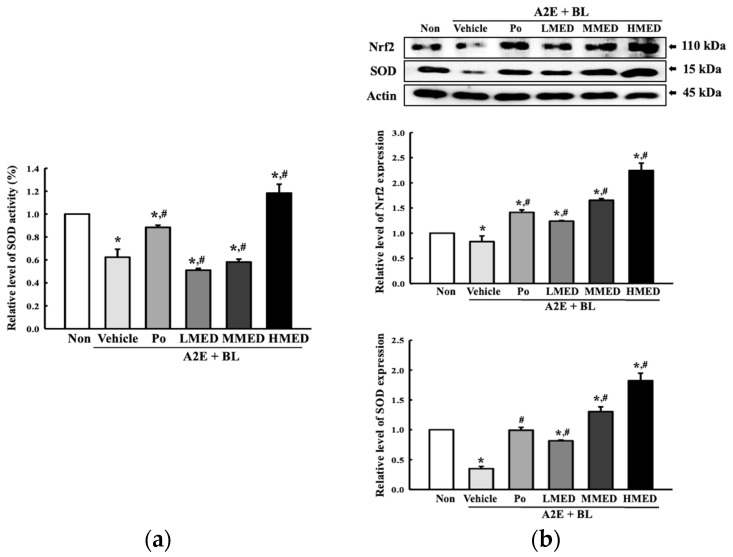
Determination of SOD and Nrf2 expression in MED + A2E + BL treated ARPE19 cells. (**a**) Measurement of SOD activity. SOD activity was measured in total cell lysate of ARPE19 cells using SOD assay kit. One SOD unit was defined as described in materials and methods. (**b**) Expression of SOD and NRF2 protein. After preparing total cell lysate from MED + A2E + BL treated ARPE19 cells, the expression level of SOD and Nrf2 protein was measured by Western blotting analysis. The total proteins homogenates were prepared from two to three samples per group, and Western blot analyses for each sample was analyzed in duplicate. All values in results are represented as the means ± standard deviation (SD). * indicated statically significance compared to the Non treated group, while ^#^ indicated statically significance compared to the Vehicle + A2E + BL treated group. Abbreviations: SOD, Superoxide dismutase; Nrf2, Nuclear factor erythroid 2–related factor 2; MED, methanol extracts of *Dipterocarpus tuberculatus* Roxb.; A2E: N-retinylidene-N-retinylethanolamine; BL, Blue light; WST-1, water-soluble tetrazolium salt-1.

**Figure 4 antioxidants-12-00329-f004:**
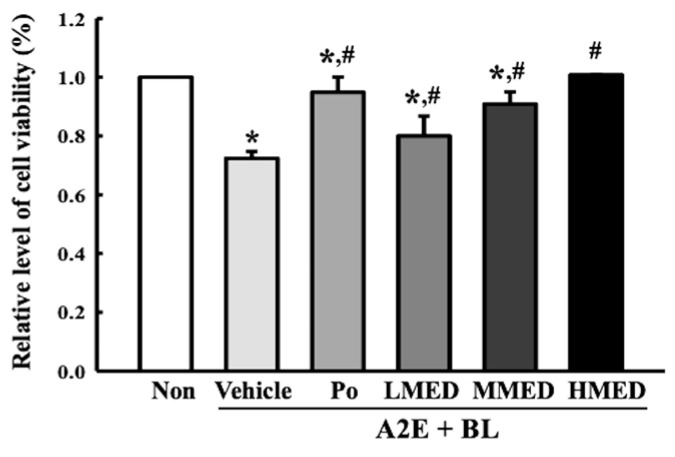
Determination of cell viability in MED + A2E + BL treated ARPE19 cells. After pretreatment of three different dosage MED for 24 h, their viability was determined using MTT assay. Two to three wells per group were used for the MTT assay, and optical density was measured in duplicates. All values in results are represented as the means ± standard deviation (SD). * indicated statically significance compared to the Non treated group, while ^#^ indicated statically significance compared to the Vehicle + A2E + BL treated group. Abbreviations: MTT, 3-(4,5-dimethylthiazol-2-yl)-2,5-diphenyltetrazolium bromide; MED, methanol extracts of *Dipterocarpus tuberculatus* Roxb.; A2E: N-retinylidene-N-retinylethanolamine; BL, Blue light.

**Figure 5 antioxidants-12-00329-f005:**
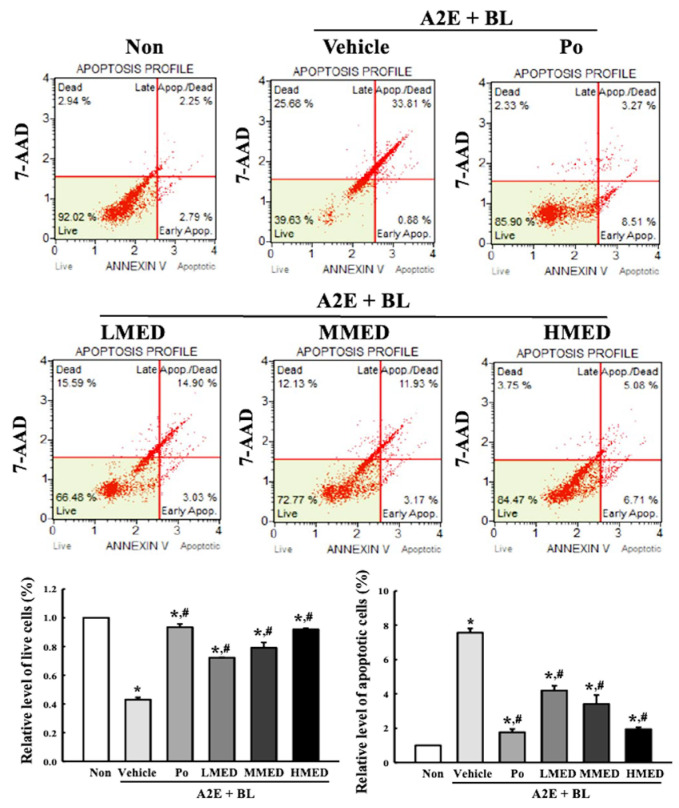
Apoptotic cell analysis in MED + A2E + BL treated ARPE19 cells. After staining with Annexin V and 7-AAD, the distribution of apoptotic cells was analyzed as described in materials and methods. All values in results are represented as the means ± standard deviation (SD). * indicated statically significance compared to the Non treated group, while ^#^ indicated statically significance compared to the Vehicle + A2E + BL treated group. Abbreviations: 7-AAD, 7-aminoactinomycin D; MED, methanol extracts of *Dipterocarpus tuberculatus* Roxb.; A2E: N-retinylidene-N-retinylethanolamine; BL, Blue light.

**Figure 6 antioxidants-12-00329-f006:**
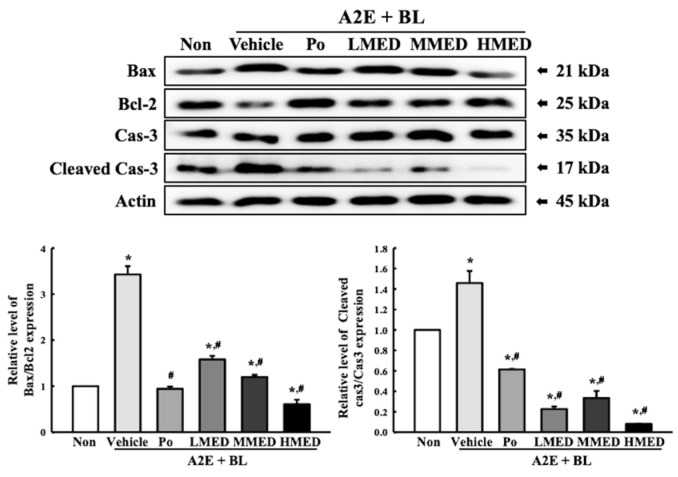
Expression of apoptotic proteins in MED + A2E + BL treated ARPE19 cells. After preparing total cell lysate from MED + A2E + BL treated ARPE19 cells, the expression level of Bax, Bcl-2, Cas-3 and Cleaved Cas-3 protein was measured by Western blotting analysis. The total proteins homogenates were prepared from two to three samples per group, and Western blot analyses for each sample was analyzed in duplicate. All values in results are represented as the means ± standard deviation (SD). * indicated statically significance compared to the Non treated group, while ^#^ indicated statically significance compared to the Vehicle + A2E + BL treated group. Abbreviations: Bax, Bcl-2-associated X protein; Bcl-2, B-cell lymphoma 2; Cas-3, Caspase-3; MED, methanol extracts of *Dipterocarpus tuberculatus* Roxb.; A2E: N-retinylidene-N-retinylethanolamine; BL, Blue light.

**Figure 7 antioxidants-12-00329-f007:**
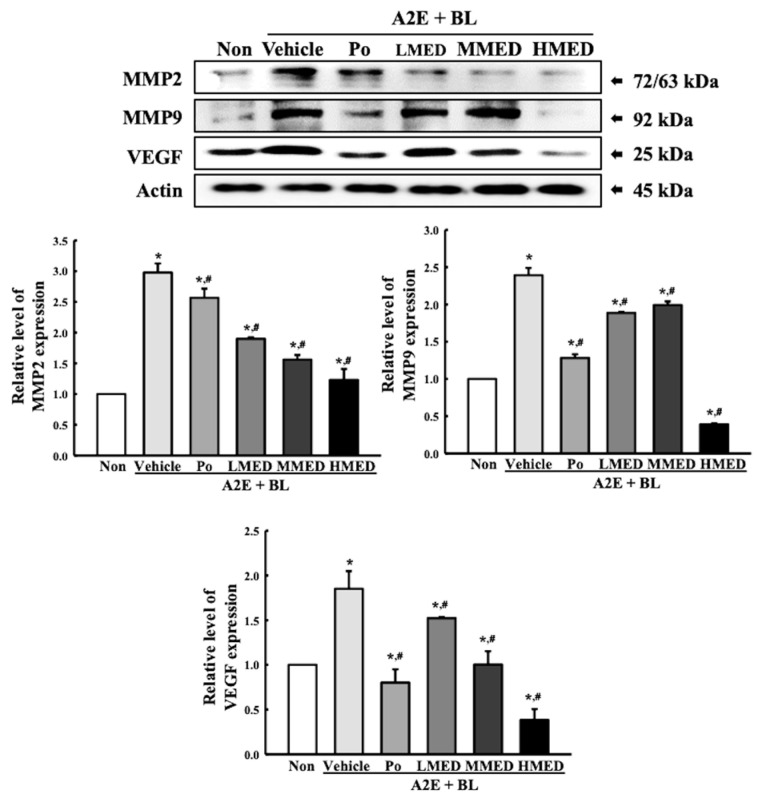
Expression of angiogenic proteins in MED + A2E + BL treated ARPE19 cells. After treatment with MED for 24 h, the expression levels of angiogenic proteins, including MMP2, MMP9, VEGF and actin were measured in total cell lysates using Western blot analysis. The total proteins homogenates were prepared from two to three samples per group, and Western blot analyses for each sample was analyzed in duplicate. All values in results are represented as the means ± standard deviation (SD). * indicated statically significance compared to the Non treated group, while ^#^ indicated statically significance compared to the Vehicle + A2E + BL treated group. Abbreviations: MMP, Matrix metalloproteinase; VEGF, Vascular endothelial growth factor; MED, methanol extracts of *Dipterocarpus tuberculatus* Roxb.; A2E: N-retinylidene-N-retinylethanolamine; BL, Blue light.

**Figure 8 antioxidants-12-00329-f008:**
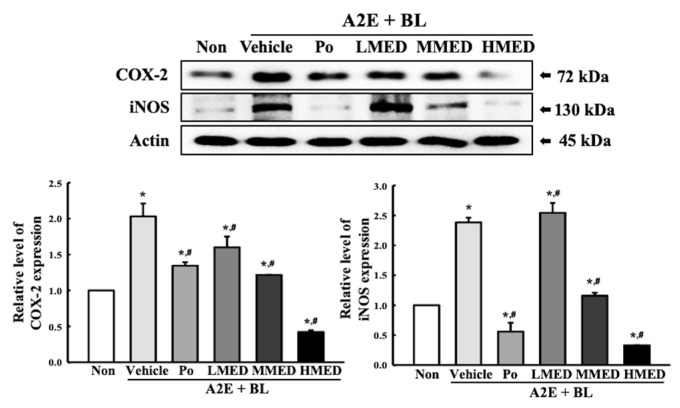
Expression of iNOS and COX-2 proteins in MED + A2E + BL treated ARPE19 cells. After treatment with MED for 24 h, the expression level of key regulators in iNOS-induced COX-2 mediated pathway, including iNOS, COX-2 and actin were measured in total cell lysates using Western blot analysis. The total proteins homogenates were prepared from two to three samples per group, and Western blot analyses for each sample was analyzed in duplicate. All values in results are represented as the means ± standard deviation (SD). * indicated statically significance compared to the Non treated group, while ^#^ indicated statically significance compared to the Vehicle + A2E + BL treated group. Abbreviations: COX-2, Cyclooxygenase-2; iNOS, Inducible nitric oxide synthase; MED, methanol extracts of *Dipterocarpus tuberculatus* Roxb.; A2E: N-retinylidene-N-retinylethanolamine; BL, Blue light.

**Figure 9 antioxidants-12-00329-f009:**
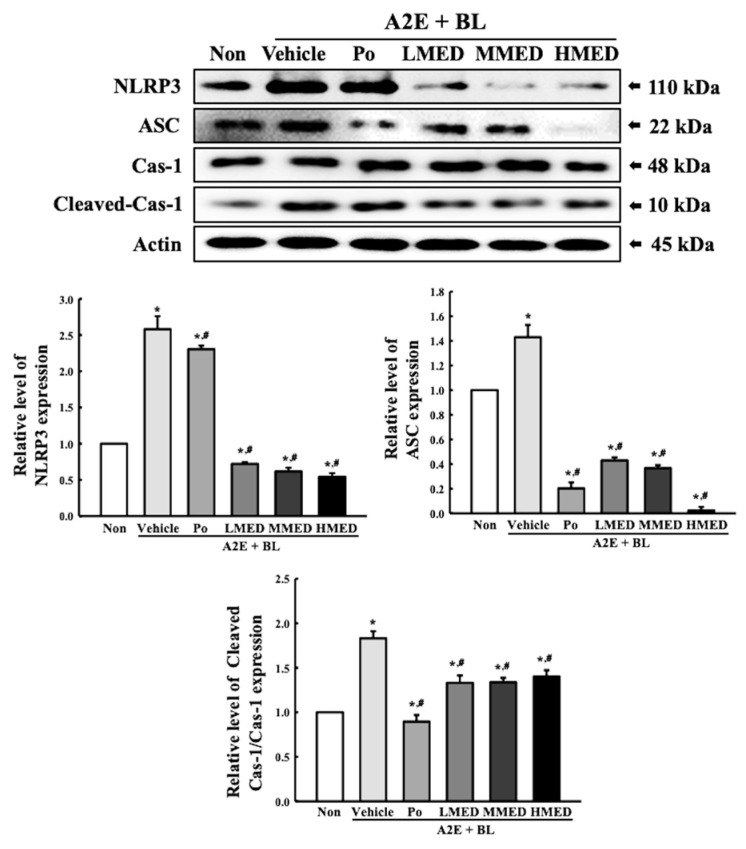
Expression of inflammasome in MED + A2E + BL treated ARPE19 cells. After treatment with MED for 24 h, the expression level of key regulators in inflammasome, including NLRP3, ASC, Cas-1, Cleaved Cas-1 and actin were measured in total cell lysates using Western blot analysis. The total proteins homogenates were prepared from two to three samples per group, and Western blot analyses for each sample was analyzed in duplicate. All values in results are represented as the means ± standard deviation (SD). * indicated statically significance compared to the Non treated group, while ^#^ indicated statically significance compared to the Vehicle + A2E + BL treated group. Abbreviations: NLRP3, NLR family pyrin domain containing 3; ASC, Apoptosis-associated speck-like protein; Cas-1, Caspase-1; MED, methanol extracts of *Dipterocarpus tuberculatus* Roxb.; A2E: N-retinylidene-N-retinylethanolamine; BL, Blue light.

**Figure 10 antioxidants-12-00329-f010:**
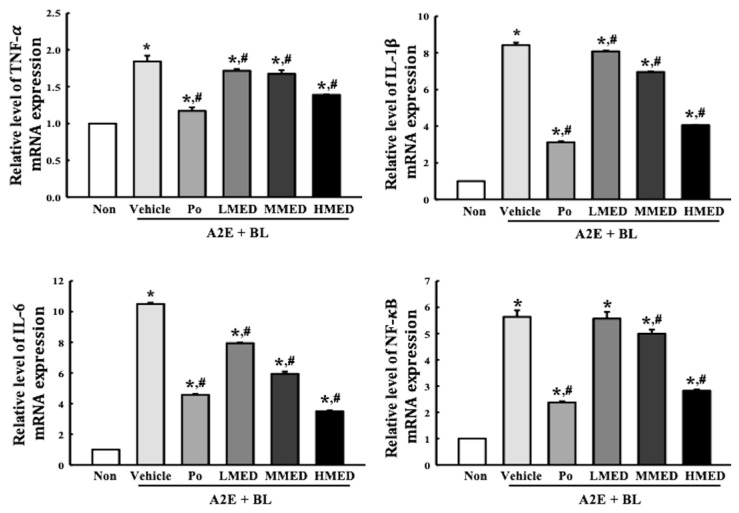
mRNA expression of inflammatory cytokines in MED + A2E + BL treated ARPE19 cells. The levels of TNF-α, IL-1β, IL-6, and NF-κB transcripts were measured in the total mRNA of ARPE19 cells by RT-qPCR using specific primers. The relative level of each transcript was determined based on the level of β-actin. The total RNAs were prepared from two to three samples per group, and RT-qPCR analyses for each sample were analyzed in duplicate. All values in results are represented as the means ± standard deviation (SD). * indicated statically significance compared to the Non treated group, while ^#^ indicated statically significance compared to the Vehicle + A2E + BL treated group. Abbreviations: TNF-α, tumor necrosis factor α; IL, Interleukin; NF-κB, Nuclear factor kappa light chain enhancer of activated B; MED, methanol extracts of *Dipterocarpus tuberculatus* Roxb.; A2E: N-retinylidene-N-retinylethanolamine; BL, Blue light.

**Figure 11 antioxidants-12-00329-f011:**
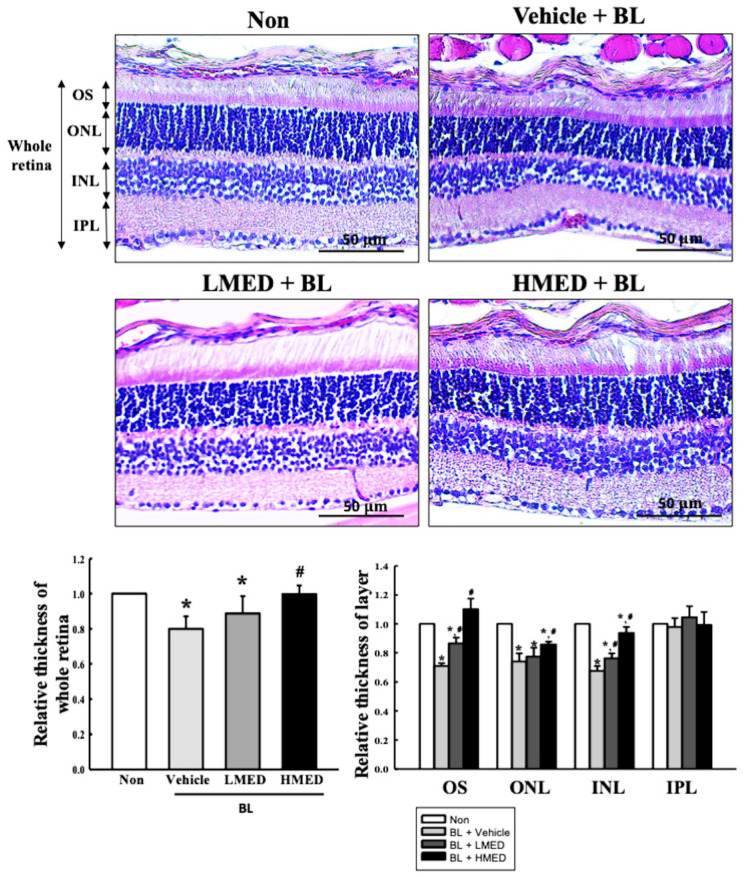
Histopathological structures in the retina of MED + BL treated Balb/c mice. H&E-stained sections of the retina were observed at 200× magnification using an optical microscope. The degree of histopathological changes in the retina tissue was measured by Image J program. The H&E-stained slides were prepared from three to five mice per group, the histopathological parameter analyses for each sample were measured in duplicate. All values in results are represented as the means ± standard deviation (SD). * indicated statically significance compared to the Non treated group, while ^#^ indicated statically significance compared to the Vehicle + A2E + BL treated group. Abbreviations: OS, Outer segment, ONL, Outer nuclear layer; INL, Inner nuclear layer; IPL, Inner plexiform layer; MED, methanol extracts of *Dipterocarpus tuberculatus* Roxb.; BL, Blue light.

**Figure 12 antioxidants-12-00329-f012:**
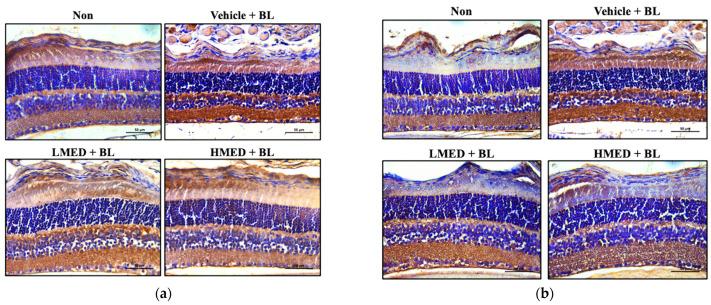
Tissue distribution of (**a**) Nrf2, (**b**) COX-2 and (**c**) iNOS protein in the retina of MED + BL treated Balb/c mice. After staining specific antibodies, the stained section of retina was observed at 200× magnification using light microscopy. Brown or dark brown indicates the expression of three proteins, and dark blue indicates the nucleus. The immunostained slides were prepared from three to five mice per group, the color density analyses for each sample were performed in duplicate. Abbreviations: Nrf2, Nuclear factor erythroid 2–related factor 2; COX-2, Cyclooxygenase-2; iNOS, Inducible nitric oxide synthase; MED, methanol extracts of *Dipterocarpus tuberculatus* Roxb.; BL, Blue light.

**Figure 13 antioxidants-12-00329-f013:**
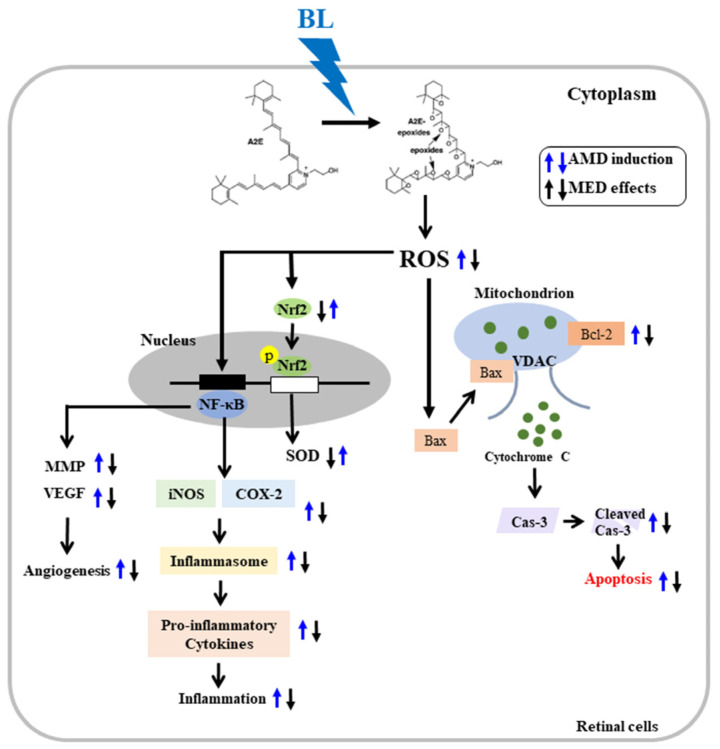
Suggested mechanism of prevention for AMD in the retinal cells. In this scheme, the treatment of MED is thought to inhibit apoptosis, angiogenesis and inflammatory response through enhancing antioxidant activity and suppressing oxidative stress retinal cells with photooxidative damages. Abbreviations: ROS, Reactive oxygen species; Nrf2, Nuclear factor erythroid 2–related factor 2; COX-2, Cyclooxygenase-2; iNOS, Inducible nitric oxide synthase; NLRP3, NLR family pyrin domain containing 3; ASC, Apoptosis-associated speck-like protein; Cas-1, Caspase-1; Bax, Bcl-2-associated X protein; Bcl-2, B-cell lymphoma 2; Cas-3, Caspase-3; BL, Blue light; MED, methanol extracts of *Dipterocarpus tuberculatus* Roxb.

## Data Availability

Data is contained within the article and Appendix A.

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
