# Peer review of "Protective Effects of Dipterocarpus tuberculatus in Blue Light-Induced Macular Degeneration in A2E-Laden ARPE19 Cells and Retina of Balb/c Mice"

_antioxidants, 2023, doi:10.3390/antiox12020329_

Round 1

Reviewer 1 Report

The manuscript reports protective effects of MED extracts in blue light induced macular degeneration. The manuscript is interesting and experiments well conducted. However the following concerns have to be addressed:

1)  Authors should report which of the Seven active components, namely, gallic acid, bergenin, ellagic acid, ε-viniferin, asiatic acid, oleanolic acid, and 2α-hydroxyursolic acid can be the main responsible of these effects. Authors find asiatic acid , hydroxyursolic acid  and ellagic acid are the principal bioactive compounds, therefore they should report control experiments to understand if the effects of these compounds is synergic or only one component is effectively responsible of the observed action.

2) Figure 13, which is the principal since summarizes observed mechanisms and hypotheses,  is not cited in the manuscript, and is not described in an organic way. 

3)  The mechanisms of action involve NRF2.Very recent literature reported how  antioxidant treatments in RPE cells activated erythroid 2-related factor Nrf2, which orchestrates the activation of cellular antioxidant pathways and ultimately inhibits apoptosis in conditions mimicking diabetic rethinopathy ( Antioxidants 2022, 11(6), 1072; https://doi.org/10.3390/antiox11061072  ) , retinitis pigmentosa (JCI Insight. 2021 Jan 25; 6(2): e145029) and in cells treated with H202 ( Biomedicines 2022, 10(2), 503; https://doi.org/10.3390/biomedicines10020503), and t-BHP (Journal of Pharmacy and Pharmacology, rgac069, https://doi.org/10.1093/jpp/rgac069)

It is important to discuss these works and compare which common mechanisms are involved in the oxidation triggered by these diverse agents other than blue light,  mimicking the onset of different eye diseases, as well as discuss which could be similarities in the mechanisms underlying the protective effect of the antioxidants.

Author Response

The manuscript reports protective effects of MED extracts in blue light induced macular degeneration. The manuscript is interesting and experiments well conducted. However the following concerns have to be addressed:

1.Authors should report which of the Seven active components, namely, gallic acid, bergenin, ellagic acid, ε-viniferin, asiatic acid, oleanolic acid, and 2α-hydroxyursolic acid can be the main responsible of these effects. Authors find asiatic acid , hydroxyursolic acid  and ellagic acid are the principal bioactive compounds, therefore they should report control experiments to understand if the effects of these compounds is synergic or only one component is effectively responsible of the observed action.

☞ According to your comments, we have further discussed above issues in Discussion section (line 578-591) as following;

“ Furthermore, in our study, asiatic acid, 2a-hydroxyursolic acid and ellagic acid are reported as the principal bioactive compounds as shown in Figure 1. They have high antioxidant activity and several biological functions even as a single compound. Asiatic acid is widely distributed in many fruits and vegetables, and has various therapeutic effects on oxidative stress, inflammation and fulminant hepatic failure [38]. Similar to asiatic acid, 2a-hydroxyursolic acid has been discovered in a variety medicinal herbs and fruits [39]. This compound shows also various pharmacological effects including antidiabetic activity, anti-inflammation, anti-obesity, anti-atherosclerosis, and anticancer [40-43,39]. Ellagic acid with high antioxidative activity is produced as second metabolites through hydrolysis of ellagitannins in various plants [44]. It has been recently received a great attention because its potential in the treatment for hu-man diseases including diabetes, cancer, cardiovascular disease, and neurodegenerative diseases [45-48]. However, we believe that the protective effects of MED against AMD may be related to the synergistic effect of three compounds rather than the effect of a single component although these effects include the effects of other unknown compounds.”

2.Figure 13, which is the principal since summarizes observed mechanisms and hypotheses,  is not cited in the manuscript, and is not described in an organic way.

☞ This is our mistake. Figure 13 has been cited in Conclusions section as following;

“Our results provided the first scientific evidences that the high antioxidant activity of MED has the potential to prevent AMD and relieve symptoms related to the condition (Figure 13).”

3.The mechanisms of action involve NRF2.Very recent literature reported how  antioxidant treatments in RPE cells activated erythroid 2-related factor Nrf2, which orchestrates the activation of cellular antioxidant pathways and ultimately inhibits apoptosis in conditions mimicking diabetic rethinopathy (Antioxidants 2022, 11(6), 1072; https://doi.org/10.3390/antiox11061072 ) , retinitis pigmentosa (JCI Insight. 2021 Jan 25; 6(2): e145029) and in cells treated with H202 (Biomedicines 2022, 10(2), 503; https://doi.org/10.3390/biomedicines10020503), and t-BHP (Journal of Pharmacy and Pharmacology, rgac069, https://doi.org/10.1093/jpp/rgac069)

It is important to discuss these works and compare which common mechanisms are involved in the oxidation triggered by these diverse agents other than blue light, mimicking the onset of different eye diseases, as well as discuss which could be similarities in the mechanisms underlying the protective effect of the antioxidants.

☞ According to your comments, we have further discussed above issues in Discussion section (line 618-640) as following;

“Meanwhile, Nrf2 protein is well known a one of important transcription factor that regulate the transcription of genes related with xenobiotics response and oxidative stress response to maintain cellular hemostasis [57]. Also, these responses involve many enzymes for detoxification including phase I, II and II drugs as well as elimination of pro-antioxidants such as glutathione synthetase (GSS), SOD and catalase [58]. Recently, the regulatory mechanism of Nrf2 has been investigated in RPE cells under mimicking the onset of different eye diseases. Nrf2 overexpression improve the morphological structure of RPE cells and survival rate in mouse model of retinal degeneration through the regulation of multiple oxidative defense pathway and glutathione pathway [59]. Furthermore, the expression level of Nrf2 increased in diabetic retinopathy condition is significantly recovered by treatment of polyunsaturated do-cosahexaenoic acid (DHA) in RPE cells [60]. A similar activation in Nrf2 signaling pathway induced in ARPE19 cells after treatment of hydrogen peroxide. This activation ameliorated with idebenone, an ubiquinone short-chain synthetic analog which have high antioxidant properties [61]. The activation of Nrf2 and transcriptional up-regulation of its downstream genes were induced by tertbutyl hydroperoxide (t-BHP)-mediated oxidative stress in human RPE cells although these responses attenuated by pretreatment of Ginkgo biloba extracts (GBE) [62]. In the present study, the expression level of Nrf2 were increased in A2E-landed ARPE19 cells after treatment of BL. Therefore, the results of the present study suggest that action mechanism of Nrf2 may considered as common mechanism involved oxidative stress response which triggered by various chemical and physical agents including hydrogen peroxide, t-BHP and BL as well as mimicking condition of eye diseases such as diabetic retinopathy and retinitis pigmentosa. Furthermore, it is believed that this mechanism of Nrf2 has the potential to be useful for evaluating the efficacy of antioxidants.

Reviewer 2 Report

The research article by Dr. Su Jin Lee et al., entitled “Protective effects of Dipterocarpus tuberculatus in blue light-induced macular degeneration in A2E-laden ARPE19 cells and retina of Balb/c mice”, investigates the protective effect of the natural compound Dipterocarpus tuberculatus (MED) against the oxidative damage which is responsible for retinal damage in age-related macular degeneration (AMD).

The study was carried out in vitro, in ARPE19 cell cultures, and in vivo, in Balb/c mice. In ARPE19 cells, treatment with toxic doses of A2E followed by exposure to blue light were carried out in order to impair cell viability. Treatment with MED was challenged in order to demonstrate its protective effect against cell damage.

The study indicates a promising compound in the treatment of AMD, which acts as a powerful antioxidant agent. Results obtained in vivo and in vitro are consistent with the conclusions. However, some critical points deserve specific Authors’ attention.

-The Authors used different wave lengths and intensity of blue light in experiments aimed at determining the effect on cell viability of increasing doses of A2E (450 nm, 2,000 lux) compared with experiments aimed at assessing the protective effects of MED on the cell death caused by BL in A2E-laden ARPE-19 cells (430 nm, 6,000 lux). Why such a discrepancy?

-Since lysosomal damage and disruption of the autophagy process in RPE are specifically involved in AMD, as also stated by the Authors, it would be worth to investigate the effects on lysosomal and autophagy status in ARPE cells by immunoblotting for specific lysosomal and autophagy antigens, such as LAMP1/2, Cathepsin D and LC3II/LC3I ratio. Similarly, in the retina of irradiated and non-irradiated mice the Authors should carry out immunostaining for Cathepsin D and LC3.

-How the Authors established the doses of MED used for in vivo experiments?

-Within the non irradiated experimental group, apart from “Non treated group”, the Authors should include the “vehicle treated group” in all the experiments.

-Figure 11 and Figure 12 do not show clearly all layers of the retina, the OS and IPL layers are not fully included in the pictures. Visualization of the OS layer is important in order to compare the effects observed in vivo with those obtained in vitro.

Author Response

The research article by Dr. Su Jin Lee et al., entitled “Protective effects of Dipterocarpus tuberculatus in blue light-induced macular degeneration in A2E-laden ARPE19 cells and retina of Balb/c mice”, investigates the protective effect of the natural compound Dipterocarpus tuberculatus (MED) against the oxidative damage which is responsible for retinal damage in age-related macular degeneration (AMD).

The study was carried out in vitro, in ARPE19 cell cultures, and in vivo, in Balb/c mice. In ARPE19 cells, treatment with toxic doses of A2E followed by exposure to blue light were carried out in order to impair cell viability. Treatment with MED was challenged in order to demonstrate its protective effect against cell damage.

The study indicates a promising compound in the treatment of AMD, which acts as a powerful antioxidant agent. Results obtained in vivo and in vitro are consistent with the conclusions. However, some critical points deserve specific Authors’ attention.

-The Authors used different wave lengths and intensity of blue light in experiments aimed at determining the effect on cell viability of increasing doses of A2E (450 nm, 2,000 lux) compared with experiments aimed at assessing the protective effects of MED on the cell death caused by BL in A2E-laden ARPE-19 cells (430 nm, 6,000 lux). Why such a discrepancy?

☞ This is our mistake. They have been corrected to the same condition.

-Since lysosomal damage and disruption of the autophagy process in RPE are specifically involved in AMD, as also stated by the Authors, it would be worth to investigate the effects on lysosomal and autophagy status in ARPE cells by immunoblotting for specific lysosomal and autophagy antigens, such as LAMP1/2, Cathepsin D and LC3II/LC3I ratio. Similarly, in the retina of irradiated and non-irradiated mice the Authors should carry out immunostaining for Cathepsin D and LC3.

☞ This is good points. Actually, further study on the effectiveness of MED on the lysosomal damage and disruption of the autophagy process will contribute to improving the value of the paper. Therefore, this issue has been added to the discussion section as a limitation and a future study of our research (line 671-676) as following;

“Moreover, the lack of any mechanism analyses in relation to lysosomal damage and dis-ruption of the autophagy process in MED+A2E+BL treated ARPE19 cells and MED+BL treated mice should be considered as a drawback of our study. Therefore, further research involving animal models will be needed on some process of cell damages to understand the mechanisms of action of MED and components.

-How the Authors established the doses of MED used for in vivo experiments?

☞ Rational basis for the dosages of MED was added into the materials and methods section (line 272-275) as following;

The dosages for MED treatment used in the AMD model was decided based on results from previous research for anti-AMD effects of natural products [32,11] and therapeutic effects of Dipterocarpus tuberculatus Roxb. [21,22].”

-Within the non irradiated experimental group, apart from “Non treated group”, the Authors should include the “vehicle treated group” in all the experiments.

☞ This is good points. Actually, we removed the vehicle treated group from main experimental groups because there was no significant difference between the Non treated group and the vehicle treated group in preliminary experiment as well as IACUC recommended that number of animals was reduced. Therefore, we have further described about above issues into materials and method section (line 266-269) as following;

“However, only Vehicle treated group were not assigned within Non treated group based on the 3R principle of PNU-IACUC to reduce the number of animals because there was no significant difference between the Non treated group and the vehicle treated group in preliminary experiment.

-Figure 11 and Figure 12 do not show clearly all layers of the retina, the OS and IPL layers are not fully included in the pictures. Visualization of the OS layer is important in order to compare the effects observed in vivo with those obtained in vitro.

☞ According to your comments, the images of Figure 11 and 12 have been replaced with the enlarged one in order to include all layers of retina.

Round 2

Reviewer 1 Report

Authors answered to all questions

Reviewer 2 Report

Though the Authors did not carry out all the changes requested, they explained the reasons for such omissions and recognized the limitation of their study. Therefore, the study was adequately improved.